# Performance analysis of dual-hop mixed RF-FSO systems combined with NOMA

**Tran Cong Hung[1], Tan N. Nguyen[2,3], N. H. K. Nhan**[3]***, Anh-Tu Le**[2], **Pham Ngoc Son[4], Thu-Ha Thi Pham[5], Miroslav Voznak[2]**

**1** Dean of School of Computer Science & Engineering, The SaiGon Internaltional University, Ho Chi Minh City, Vietnam, **2** Faculty of Electrical Engineering and Computer Science, VSB-Technical University of Ostrava, Ostrava, Czechia, **3** Communication and Signal Processing Research Group, Faculty of Electrical and Electronics Engineering, Ton Duc Thang University, Ho Chi Minh City, Vietnam, **4** Faculty of Electrical and Electronics Engineering, Ho Chi Minh City University of Technology and Education, Thu Duc City, Ho Chi Minh City, Vietnam, **5** Faculty of Electrical and Electronics Engineering, Ton Duc Thang University, Ho Chi Minh City, Vietnam

* nguyenhuukhanhnhan@tdtu.edu.vn

**Data Availability Statement:** All relevant data are within the manuscript and its Supporting information files.

## Abstract

This paper investigates the performance of hybrid radio frequency/free space optical (RF/FSO) systems combined with non-orthogonal multiple access communications technology. We examine a scenario where the source and destination are separated by a large distance, with no direct link between them. The relay, denoted R, operates using the decode-and-forward (DF) protocol. Under the DF relaying scheme, the relay employs successive interference cancellation (SIC). In this setup, the FSO link from the source to the relay follows a Gamma-Gamma distribution, while the RF link from the relay to multiple users follow a Nakagami-$m$ distribution. Based on this system model, we analyze the outage probability (OP). Our findings indicate a direct relationship between SIC and OP performance: the higher the SIC capability, the more effective the system. In addition, the system's performance is dependent on the parameters of the FSO channel. Finally, Monte Carlo simulations are presented to further validate our framework and findings.

## 1 Introduction

One of the most significant breakthroughs in the history of telecommunications was the invention of wireless communication. The proliferation and prevalence of wireless devices have exceeded expectations from a few decades ago and are predicted to continue growing at an exponential rate. Radio frequency (RF) technology has dominated wireless communication systems, occupying both licensed and unlicensed spectrums, in a wide variety of contexts [1, 2]. This dominance has led to widespread adoption and the existence of a large number of RF devices. However, several issues underlie the use of the RF band, including limited capacity, high costs of licensed spectrum technologies, and interference from unlicensed spectrum technologies. With the number of consumer devices growing exponentially, researchers are exploring novel wireless communication methods. It is essential to evaluate additional frequency bands in the electromagnetic spectrum for data transfer since the demand for constant

**Funding:** This research is funded by the European Union within the REFRESH project - Research Excellence For Region Sustainability and High-tech Industries ID No. CZ.10.03.01/00/22 003/0000048 of the European Just Transition Fund and by the Ministry of Education, Youth and Sports of the Czech Republic (MEYS CZ) through the project SGS ID No. SP 061/2024 conducted by VSB - Technical University of Ostrava.

**Competing interests:** The authors have declared that no competing interests exist.

communication availability exceeds the RF spectrum's capacity. In this context, free-space optical (FSO) communication systems have drawn much interest from both academic and industry communities for their ease of implementation, low power consumption, high data rates and bandwidth, and free spectrum licensing [3–7]. Compared to traditional RF communications, FSO communications offer higher throughput, the use of unlicensed spectrums, cost-effectiveness, and other advantages. FSO systems represent a promising solution for various communications scenarios, which include backhaul/fronthaul link for cellular systems, aerial/drone-assisted wireless emergency communications for disaster recovery, and optical communications for space exploration [6, 8, 10].

Since mixed FSO-RF networks can benefit from both the broadcasting nature and wide coverage area of the RF technique as well as the high channel capacity, license-free operation, and enhanced security of the FSO technique, they have attracted a lot of interest. In an FSO-RF system, a cooperative relay is used as an intermediary to transfer signals from the transmitter to the receiver. Relay strategies are divided into two categories: protocols for amplify-and-forward (AF) relaying and decode-and-forward (DF) relaying [11]. Various fading models are also employed in the FSO and RF channels of hybrid FSO-RF relaying systems. The FSO link can function according to Gamma-Gamma [12–16] or exponentiated Weibull distributions [17, 18], among others. In contrast, the RF link can operate under Rayleigh [12, 13], Nakagami-$m$ [14, 17, 19], generalized $K$–$\mu$ [18], and K distributions [20]. For example, the hybrid FSO-RF system described in [12], applies RF and FSO link defined according to the Rayleigh and Gamma-Gamma distributions, respectively, and uses an AF relay. In another study [17], both fixed-gain and variable-gain relay schemes are examined, using an RF link modeled according to the Nakagami-$m$ distribution and an FSO link under exponentiated Weibull fading. A dual-hop hybrid FSO-RF system characterized by RF and FSO link that follow Nakagami-$m$ and double generalized Gamma distributions, respectively, is described in [19]. Similarly, an asymmetric dual-hop AF relaying system employing RF and FSO link under Nakagami-$m$ and Gamma-Gamma fading, respectively, is discussed in [14].

Non-orthogonal multiple access (NOMA) also leverages non-orthogonal resource allocation to enable a high number of users access the network, in contrast to traditional orthogonal multiple access (OMA) technologies. Recently, NOMA has garnered much interest as an essential technology for 5G wireless communication networks [21–25]. To ensure fairness among users, NOMA assigns lower power levels to users with stronger channel gains and higher power levels to users with weaker channel gains [26, 27]. These signals are then superposed at the source. Academic and industrial research have also demonstrated that NOMA can effectively support vast connectivity in addition to its increased spectral efficiency. This capability is crucial for supporting the Internet of Things (IoT) features in the forthcoming 5G Next networks [28–30]. Due to NOMA's interoperability with various communication technologies, it can be seamlessly integrated into both current and future wireless systems. For instance, it has been demonstrated that NOMA is compatible with traditional OMA methods such as orthogonal frequency division multiple access (OFDMA) and time division multiple access (TDMA) [31]. NOMA has also been incorporated into the next digital TV standard, known as layered division multiplexing [32], which enhances the spectral efficiency of TV broadcasting by superimposing numerous data streams. Recent research has also explored the use of NOMA with unmanned aerial vehicle (UAV) systems [33, 34] and in satellite architectures [35, 36]. The application of multiple intelligent reflecting surfaces (IRSs) with distinct phase shifts to support NOMA networks has also been examined [37, 38]. These aforementioned instances highlight NOMA's enormous potential not only for 5G networks but also for various current and future wireless systems.

## 1.1 Related work

Exploiting the advantages of both the high data rates facilitated by FSO systems and the high spectral efficiency introduced by NOMA serves as the primary motivation for combining FSO and NOMA. Another reason is that NOMA typically operates well in situations where the power received from various transmitters varies significantly [27]. This condition is often encountered in FSO systems because adverse weather conditions, such as haze and fog, result in higher path loss for FSO link compared to RF lines. However, the combination of FSO and NOMA has only recently been explored in the literature [39–41]. In [39], the authors consider employing NOMA for the FSO backhauling system, which consists of two base stations and one central unit. Another study [40] examines a multi point-to-point system consisting of K users and a central node using NOMA over an FSO channel. In [41], the authors investigate a low-altitude platform (LAP)-aided dual-hop relaying system that combines NOMA and FSO communications technology.

In addition, the incorporation of NOMA with mixed FSO-RF networks to achieve better communication performance [42–47]. A two-hop uplink NOMA RF-FSO network was investigated in [42] in order to meet the high throughput requirements for the backhaul connection. The closed-form formulation of OP and ergodic channel capacity under various interference situations was also provided. The subject of the combined optimization of power allocation and harvesting time was further investigated in [43] when the analytical expressions of OP and throughput were taken into consideration in the context of an energy harvesting enabled mixed FSO-RF network. By including NOMA into both the FSO and RF connections, the authors of [44] suggested a down-link FSO-RF network and determined the closed-form expression of OP for every user. The analytical formulation of OP was introduced in [45], taking into account that the RF channel has a Rayleigh distribution and the FSO channel has an M-distribution. [46] provided the analytical and asymptotic formulas of secrecy OP (SOP) for an FSORF system with NOMA while accounting for the imperfect channel state information. For a NOMA-based hybrid RF-UWOC system, the asymptotic formulations of ergodic capacity and the closed-form formulas of OP were provided in [47].

## 1.2 Contributions

Based on the above-mentioned issues, it is highly intriguing and promising to research the FSO-RF cooperation with NOMA. In this paper, we consider a data transmission protocol where a relay is available between the source and multiple destinations to investigate the effect of OP. Table 1 shows the comparison between our work and related papers and the key notations are used in this paper, as shown in Table 2. Our main contributions are as follows:

- A proposed dual-hop mixed RF/FSO system that includes both RF and FSO link. In this system, the FSO link, which connects the source to the relay, follows a doubly generalized

**Table 1. The comparison of our work and the previous works.**

| Ref./Prop. | Dual-hop FSO-RF | Nakagami-$m$ fading | NOMA | OP | Asymptotic | Stochastic Geometry |
|---|---|---|---|---|---|---|
| [48] | ✓ | ✓ | ✓ | ✓ | X | X |
| [49] | ✓ | ✓ | ✓ | ✓ | ✓ | X |
| [50] | ✓ | ✓ | X | ✓ | X | X |
| [51] | X | X | X | ✓ | X | ✓ |
| [52] | ✓ | X | X | ✓ | ✓ | X |
| Our study | ✓ | ✓ | ✓ | ✓ | ✓ | ✓ |

**Table 2. Main parameter notation.**

| Symbol | Notation | Symbol | Notation |
|---|---|---|---|
| $x_i$ | Signal at User $i$, $i \in \{1, 2\}$ | $a_i$ | Power allocation coefficients at User $i$ |
| $R_{th}^i$ | Target rate at User $i$ | $\gamma_{th}^i$ | Threshold rate of User $i$ |
| $P_S$ | Transmit power at Source | $P_R$ | Transmit power at Relay |
| $d_1$ | Distance from $R$ to near user | $d_2$ | Distance from $R$ to far user |
| $L$ | Distance from $S$ to $R$ | $\delta$ | Path loss exponent |
| $h_0$ | Channel gain from $S$ to $R$ | $h_1$ | Channel gain from $R$ to near user |
| $h_2$ | Channel gain from $R$ to far user | $\mathcal{CN}(u, \sigma^2)$ | Complex Gaussian random variable |
| $f_X(\cdot)$ | Probability density function (PDF) of $X$ | $F_X(\cdot)$ | Cumulative distribution function (CDF) of $X$ |
| $\Pr(\cdot)$ | Probability operator | $\mathbb{E}\{\cdot\}$ | Expectation operator |
| $|\cdot|$ | Absolute value of a complex number | $\Gamma(\cdot)$ | Gamma function |
| $\Gamma(\cdot, \cdot)$ | Upper incomplete Gamma function | $\gamma(\cdot, \cdot)$ | Lowe incomplete Gamma function |
| $K_q(\cdot)$ | Bessel function of the second kind | $G_{p,q}^{m,n}(.)$ | Meijer G-function |

Gamma distribution. The RF link, connecting the relay to multiple users, follows the Naka-gami-$m$ distribution. The cooperative relay, functioning as an intermediary to transmit signals from the transmitter to the receiver, employs DF relaying. We examine the signal-to-interference-plus-noise ratio (SINR) for both the FSO and RF link when relay R uses successive interference cancellation (SIC), in addition to the SINR when relay R applies imperfect SIC (ipSIC).

- To ensure the secure performance of the proposed system, we derive the OP expressions for both near and far users. Notably, we investigate the outages in the high-SNR regime. Analysis and simulation results evaluate the impacts of key parameters on performance.

## 1.3 Overview

Following the introduction in Section 1, Section 2 presents the system model of FSO/RF cooperative NOMA and the channel characteristics of FSO and RF. In Section 3, we obtain the statistical properties of the FSO/RF link. In Section 4, we derive the analytical expressions for the OP for three cases: near users, far users, and both near and far users under a high SNR regime. Section 5 provides numerical and simulation results to verify the theoretical analyses. Section 6 concludes the paper with a summary of its main findings and contributions.

## 2 System model and channel characteristics

In this paper, we consier the FSO/RF Cooperative NOMA system model with a source ($S$), a relay ($R$), and two users as in Fig 1. It is assumed that $R$ operates in half-duplex mode and that the channel state information (CSI) is perfect at each terminal to detect the signal. The communications from $S$ to $R$ and $R$ to $D_i$, $i \in \{1, 2\}$ therefore occupy two-time slots. In the first time slot, by applying NOMA principle, $S$ using NOMA signaling $\sqrt{a_1 P_S} x_1 + \sqrt{a_2 P_S} x_2$ to transmits to $R$ through the FSO link, where $P_S$ is the transmission power at $S$, $x_1$ and $x_2$ are the unit-power data symbols, $a_1$ and $a_2$, $a_2 > a_1$ are the corresponding power allocation coefficients. The signal received at the $R$ is therefore expressed as

$$Y_R = h_0(\sqrt{a_1 P_S} x_1 + \sqrt{a_2 P_S} x_2) + n_R, \tag{1}$$

where $n_R$ is the additive Gaussian noise at the relay in which $n_i \sim \mathcal{CN}(0, \sigma^2)$.

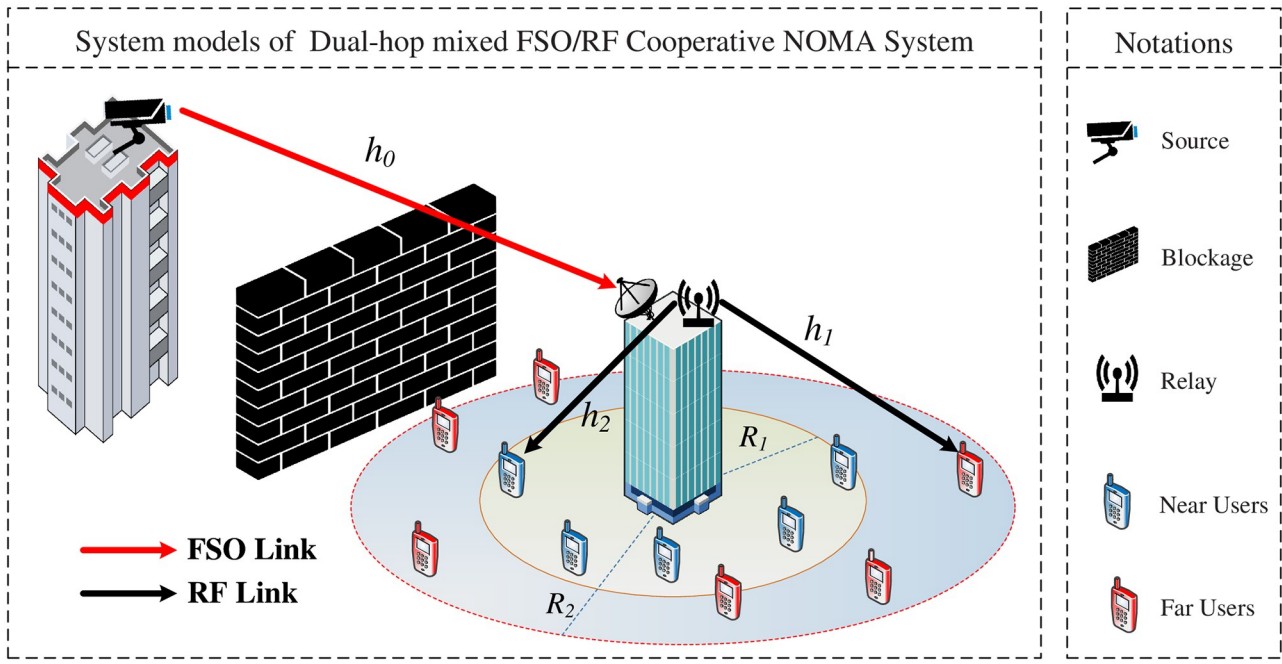

**Fig 1. FSO/RF cooperative NOMA system model.**

## 2.1 FSO link

Using the NOMA protocol, $R$ decodes the information from $S$ in $x_2$ by considering $x_1$ as an interference signal. The received SINR to identify $x_2$ at $R$ is therefore given by

$$
\begin{aligned}
\gamma_{R,x_2}^{SIC} &= \frac{a_2 P_S |h_0|^2}{a_1 P_S |h_0|^2 + \sigma^2} \\
&= \frac{a_2 \rho |h_0|^2}{a_1 \rho |h_0|^2 + 1},
\end{aligned}
\tag{2}
$$

where $\rho = P_S/\sigma^2$ denotes the transmit signal-to-noise ratio (SNR). Note that $x_1$ and $x_2$ are normalized unity power signals, i.e, $\mathbb{E}\{x_1^2\} = \mathbb{E}\{x_2^2\} = 1$.

After performing SIC at $R$ to identify $x_2$, the received SINR at $R$ to detect $x_1$ in the case of imperfect SIC is expressed as

$$
\gamma_{R,x_1}^{ipSIC} = \frac{a_1 \rho |h_0|^2}{\chi a_2 \rho |h_0|^2 + 1}.
\tag{3}
$$

where $\chi$, $0 \leq \chi \leq 1$, representing the efficiency of SIC for $x_2$ at $R$. The cases $\chi = 0$ and $\chi = 1$ correspond to perfect SIC (pSIC) and imperfect SIC (ipSIC) in [53], respectively.

## 2.2 RF link

After receiving $Y_R$, the relay decodes and transmits the signal to end users via Nakagami-$m$ fading RF channel. By similarly applying NOMA principle in FSO link, $R$ transmit the NOMA signaling $\sqrt{a_1 P_R} x_1 + \sqrt{a_2 P_R} x_2$ to user $D_i$. Thus, the signals received at $D_i$ is therefore expressed

as

$$Y_{D_u} = \frac{h_u}{\sqrt{d_u^\delta}} \left( \sqrt{a_1 P_R} x_1 + \sqrt{a_2 P_R} x_2 \right) + n_u \quad , u \in \{1, 2\} \tag{4}$$

where $P_R$ is the normalized transmission power at $R$, $d_u$ is the distance between $R$ and $D_u$, and $\delta$ is the path-loss exponent.

Using the NOMA protocol, since the user with the highest received signal strength, $D_1$, applies SIC, the SINRs required to perform SIC for $x_1$ and $x_2$ at $D_1$ are given by

$$\gamma_{1,x_2}^{SIC} = \frac{a_2 \rho |h_1|^2}{a_1 \rho |h_1|^2 + d_1^\delta}, \tag{5}$$

$$\gamma_{1,x_1}^{ipSIC} = \frac{a_1 \rho |h_1|^2}{\chi a_2 \rho |h_1|^2 + d_1^\delta}. \tag{6}$$

In contrast, $D_2$ decodes its intended signal $x_2$ by treating $x_1$ as interference. Consequently, the SINR at $D_2$ is

$$\gamma_{2,x_2} = \frac{a_2 \rho |h_2|^2}{a_1 \rho |h_2|^2 + d_2^\delta}. \tag{7}$$

## 3 Performance analysis

In an investigation of the system performance, we first obtain the statistical property of the FSO/RF link.

### 3.1 Channel model

**3.1.1 Fading statistics of the FSO channel.** Let $\gamma_0 \triangleq |h_0|^2$ be a $K_G$ distributed random variables (RV) with three gamma-gamma parameters that were obtained from its square. This PDF is provided by [54, Eq. (1)]:

$$f_{\gamma_0}(x; \alpha, \beta, \Omega_0) = \frac{2(\alpha\beta)^{\frac{\alpha+\beta}{2}} x^{\frac{\alpha+\beta}{2}-1}}{\Gamma(\alpha)\Gamma(\beta)\Omega_0^{\frac{\alpha+\beta}{2}}} K_{\alpha-\beta}\left( 2\sqrt{\frac{\alpha\beta}{\Omega_0} x} \right), \tag{8}$$

where $\alpha \geq 0$ and $\beta \geq 0$ are the factors that determine the shaping of the small-scale and large-scale eddies in the scattering environment, and $\Omega_0$ relates to the mean as $\mathbb{E}[\gamma_0] = \Omega_0$, in which $\mathbb{E}[\cdot]$ denotes expectation. The two effective numbers, $\alpha$ and $\beta$, relate to the atmospheric conditions and are expressed as [55]

$$\alpha = \left( \exp\left[ \frac{0.49\sigma_0^2}{(1 + 0.18d_0^2 + 0.56\sigma_0^{12/5})^{7/6}} \right] - 1 \right)^{-1}, \tag{9}$$

$$\beta = \left( \exp\left[ \frac{0.51\sigma_0^2(1 + 0.69\sigma_0^{12/5})^{-5/6}}{(1 + 0.9d_0^2 + 0.62d_0^2\sigma_0^{12/5})^{5/6}} \right] - 1 \right)^{-1}, \tag{10}$$

where $\sigma_0^2 = 0.492 C_n^2 \hat{k}^{7/6} L^{11/6}$ denote the Rytov variance, $d_0 = \sqrt{2\pi D^2/4L\lambda}$, $L$ is the distance between $S$ and $R$, $\lambda$ is the operational wavelength, $D$ is the aperture diameter of the receiver,

and $C_n^2$ is the altitude-dependent index of the refractive structure parameter determining the turbulence strength. Additionally, we presume that $C_n^2$ is constant during reasonably long transmit bit intervals [56].

Since it represents a variety of models often used in communication systems for a variety of combinations of $\alpha$ and $\beta$, the distribution in (8) is general. Thus, as $\alpha \to \infty$, it approximates the well-known Gamma distribution (or alternatively squared Nakagami-$m$ [57]), whereas for $\beta = 1$, it corresponds to the statistics of a squared $K$-distributed RV, with a PDF given by

$$f_{\gamma_0}(x; \alpha, 1, \Omega_0) = \frac{2(\alpha)^{\frac{\alpha+\beta}{2}} x^{\frac{\alpha+\beta}{2}-1}}{\Gamma(\alpha)\Omega_0^{\frac{\alpha+\beta}{2}}} K_{\alpha-1}\left(2\sqrt{\frac{\alpha}{\Omega_0}x}\right), \tag{11}$$

where $K_q(\cdot)$ denotes the $q^{th}$ order of the modified Bessel function of the second kind.

It can then be simplified to the power statistics of the double Rayleigh model, which is often employed in cascade multipath fading channels, with the PDF obtained from [58] for the particular case of $\alpha$ and $\beta$:

$$f_{\gamma_0}(x; 1, 1, \Omega_0) = \frac{2}{\Omega_0} K_0\left(2\sqrt{\frac{x}{\Omega_0}}\right). \tag{12}$$

Now, we obtain the $p$-th moment of $\gamma_0$ as given by [54]:

$$\mathbb{E}[\gamma_0^p] = \xi^{-p}\frac{\Gamma(\alpha+p)\Gamma(\beta+p)}{(\alpha\beta)^p\Gamma(\alpha)\Gamma(\beta)} \tag{13}$$

where $\xi = \frac{\alpha\beta}{\Omega_0}$. Furthermore, [59, Eq. (7)] and [60, Eq. (9.31.5)] can be applied to define the cumulative density function (CDF), expressed as

$$F_{\gamma_0}(x; \alpha, \beta, \Omega_0) = \frac{1}{\Gamma(\alpha)\Gamma(\beta)} G_{1,3}^{2,1}\left(\xi x \left|\begin{array}{c} 1 \\ \alpha, \beta, 0 \end{array}\right.\right), \tag{14}$$

where $G_{p,q}^{m,n}(.)$ is the Meijer G-function [20, Eq. 9.301].

## 3.2 Fading statistics of the RF channel

Both the $R$ to $D_1$ and $R$ to $D_2$ routes in the RF path are modeled with Nakagami-$m$ distribution and fading severity parameters $m_{h_1}$ and $m_{h_2}$, respectively. The CDF and PDF of $|h_u|^2$, $u \in \{1, 2\}$ may therefore be expressed as [61]

$$f_{|h_u|^2}(x) = \frac{\mu_u{}^{m_u} e^{-\mu_u x} x^{m_u-1}}{\Gamma(m_u)}, \tag{15a}$$

$$F_{|h_u|^2}(x) = 1 - e^{-\mu_u x}\sum_{s=0}^{m_u-1}\frac{\mu_u^s x^s}{s!}, \tag{15b}$$

where $\mu_u = \frac{m_u}{\lambda_u}$, in which $m_u$ and $\lambda_u$ represent the integer fading factor and the mean, respectively.

In addition, the scenario $R_2 \geq R_1$ ($R_1$ and $R_2$ are the radii of the inner and outer circles, respectively), note that the users are deployed in $D_1$ and $D_2$ based on homogeneous poisson point processes. Consequently, NOMA users are characterized as independently and identically distributed (i.i.d.) points in $D_1$ and $D_2$, indicated by $d_u$, $u \in \{1, 2\}$, which carry location information about nearby and distant users, respectively. The $d_1$ and $d_2$ probability density

functions (PDFs) are provided by [62]:

$$f_{d_1}(x) = \frac{\partial}{\partial x} \frac{\pi x^2}{\pi R_1^2} = \frac{2x}{R_1^2},$$ (16)

$$f_{d_2}(x) = \frac{\partial}{\partial x} \frac{\pi(x^2 - R_1^2)}{\pi(R_2^2 - R_1^2)} = \frac{2x}{R_2^2 - R_1^2}.$$ (17)

# 4 Outage probability analysis

## 4.1 Outage probability of $D_1$

Signal $x_1$ cannot be received successfully by $D_1$ when the following cases occur: when $R$ is unable to decode either $x_2$ or $x_1$ and when $D_1$ fails to perform SIC for $x_2$ and is thus unable to decode $x_1$.

The outage probabilities in the first and second cases are given by

$$\mathcal{P}_{out}^1 = 1 - \Pr\begin{pmatrix} \gamma_{R,x_2}^{SIC} > \gamma_{th}^2, \gamma_{R,x_1}^{ipSIC} > \gamma_{th}^1, \\ \gamma_{1,x_2}^{SIC} > \gamma_{th}^2, \gamma_{1,x_1}^{ipSIC} > \gamma_{th}^1 \end{pmatrix}$$

$$= 1 - \mathcal{I}_1 \mathcal{I}_2,$$ (18)

where $\mathcal{I}_1 = \Pr(\gamma_{R,x_2}^{SIC} > \gamma_{th}^2, \gamma_{R,x_1}^{ipSIC} > \gamma_{th}^1)$, $\mathcal{I}_2 = \Pr(\gamma_{1,x_2}^{SIC} > \gamma_{th}^2, \gamma_{1,x_1}^{ipSIC} > \gamma_{th}^1)$ and $\gamma_{th}^u = 2^{2R_{th}^u} - 1$, with $R_{th}^u$ being the target rate at of $D_u$, $\forall u \in \{1, 2\}$.

The closed-form formulation of $\mathcal{P}_{out}^1$ is provided in Proposition 1 from (18).

**Proposition 1** *The proposed FSO/RF dual-hop relay system's $\mathcal{P}_{out}^1$ is calculated from the Eq:*

$$\mathcal{P}_{out}^1 = 1 - 2\sum_{s=0}^{m_1-1} \frac{\mu_1^s \Theta_{max}^s \gamma(\vartheta, \mu_1 \Theta_{max} R_1^\delta)}{s! R_1^2 \delta \mu_1^\vartheta \Theta_{max}^\vartheta}$$

$$\times \left(1 - \frac{1}{\Gamma(\alpha)\Gamma(\beta)} G_{1,3}^{2,1}\left(\xi\Theta_{max} \bigg| \begin{matrix} 1 \\ \alpha, \beta, 0 \end{matrix}\right)\right),$$ (19)

*where* $\Theta_1 = \frac{\gamma_{th}^1}{\rho(a_1 - \chi a_2 \gamma_{th}^1)}$, $\Theta_2 = \frac{\gamma_{th}^2}{\rho(a_2 - a_1 \gamma_{th}^2)}$, $\Theta_{max} = \max(\Theta_1, \Theta_2)$ *and* $\vartheta = \frac{\delta s + 2}{\delta}$.

**Proof 1** *See* S1 Appendix.

## 4.2 Outage probability of $D_2$

Similarly, in the first case, the signals $x_1$ and $x_2$ cannot be effectively received by $R$. Then, in the second case, $D_2$ fails to decode the signal $x_2$. The outage probability of $D_2$ is therefore be calculated as follows:

$$\mathcal{P}_{out}^2 = 1 - \Pr(\gamma_{R,x_2}^{SIC} > \gamma_{th}^2, \gamma_{R,x_1}^{ipSIC} > \gamma_{th}^1)$$

$$+ \Pr(\gamma_{R,x_2}^{SIC} > \gamma_{th}^2, \gamma_{R,x_1}^{ipSIC} > \gamma_{th}^1)\Pr(\gamma_{2,x_2}^{SIC} < \gamma_{th}^2)$$

$$= 1 - \mathcal{I}_1 + \mathcal{I}_1 \mathcal{I}_3,$$ (20)

where $\mathcal{I}_3 = \Pr(\gamma_{2,x_2} < \gamma_{th}^2)$.

Proposition 2 gives the closed-form expression of $\mathcal{P}_{out}^2$, derived from (20).

**Proposition 2** *The proposed FSO/RF dual-hop relay system's* $\mathcal{P}_{out}^2$ *is computed from the Eq:*

$$
\begin{aligned}
\mathcal{P}_{out}^2 \quad &= 1 - \left[ 1 - \frac{1}{\Gamma(\alpha)\Gamma(\beta)} G_{1,3}^{2,1}\left( \xi\Theta_{\max} \middle| \begin{matrix} 1 \\ \alpha, \beta, 0 \end{matrix} \right) \right] \\
&+ \left[ 1 - \frac{1}{\Gamma(\alpha)\Gamma(\beta)} G_{1,3}^{2,1}\left( \xi\Theta_{\max} \middle| \begin{matrix} 1 \\ \alpha, \beta, 0 \end{matrix} \right) \right] \\
&\times \left\langle 1 - \frac{2}{(R_2^2 - R_1^2)} \sum_{s=0}^{m_2-1} \frac{\mu_2^s \Theta_2^s}{s! \delta \mu_2^\vartheta \Theta_2^\vartheta} \right. \\
&\times [\gamma(\vartheta, \mu_2\Theta_2) + \Gamma(\vartheta, \mu_2\Theta_2) \\
&\left. - \gamma(\vartheta, \mu_2\Theta_2 R_1^\delta) - \Gamma(\vartheta, \mu_2\Theta_2 R_2^\delta)] \right\rangle,
\end{aligned}
\tag{21}
$$

*where* $\Theta_2$, $\Theta_{\max}$, $\xi$ *and* $\vartheta$ *are announced previously.*

**Proof 2** *See* S2 Appendix.

## 4.3 High SNR regime

In this section, we examine outages in the high SNR regime, i.e., when $\rho \rightarrow \infty$.

From [63, Eq. (18)], the $G_{p,q}^{m,n}(.)$ Meijer G-function is given by the approximation:

$$
\begin{aligned}
& G_{p,q}^{m,n}\left( z \middle| \begin{matrix} a_1, \ldots, a_n, a_{n=1}, \ldots, a_p \\ b_1, \ldots, b_m, b_{m+1}, \ldots, b_q \end{matrix} \right) \\
& \approx \sum_{k=1}^{m} \frac{\prod_{j=1, j\neq k}^{m} \Gamma(b_j - b_k) \prod_{j=1}^{n} \Gamma(1 - a_j + b_k)}{\prod_{j=n+1}^{p} \Gamma(a_j - b_k) \prod_{j=m+1}^{q} \Gamma(1 - b_j + b_k)} z^{b_k}.
\end{aligned}
\tag{22}
$$

Then, we obtain $\mathcal{I}_1^\infty$ from:

$$
\begin{aligned}
\mathcal{I}_1^\infty \quad &\approx 1 - \frac{1}{\Gamma(\alpha)\Gamma(\beta)} \\
&\times \left[ \frac{\Gamma(\beta - \alpha)}{\alpha} (\xi\Theta_{\max})^\alpha + \frac{\Gamma(\alpha - \beta)}{\beta} (\xi\Theta_{\max})^\beta \right].
\end{aligned}
\tag{23}
$$

When $x \rightarrow 0$, the approximate expressions of the CDF for the channel gain $|h_u|^2$, $u \in \{1, 2\}$ are given by [64, Eq. (16a)]

$$
F_{|h_u|^2}(x) \approx \frac{(\mu_u x)^{m_u}}{m_u!}.
\tag{24}
$$

Aided by (24), we obtain $\mathcal{I}_2^\infty$ and $\mathcal{I}_3^\infty$:

$$
\begin{aligned}
\mathcal{I}_2^\infty &= \int_0^{R_1} f_{d_1}(x)\left[1 - F_{|h_1|^2}(\Theta_{\max} x^\delta)\right] dx \\[2mm]
&= \frac{2}{R_1^2} \int_0^{R_1} x\left[1 - \frac{(\mu_1 \Theta_{\max})^{m_1} x^{m_1 \delta}}{m_1!}\right] dx \\[2mm]
&= 1 - 2\frac{\mu_1^{m_1} \Theta_{\max}^{m_1} R_1^{m_1 \delta}}{m_1!(m_1 \delta + 2)},
\end{aligned}
\tag{25}
$$

and

$$
\begin{aligned}
\mathcal{I}_3^\infty &= \int_{R_1}^{R_2} f_{d_2}(x)[1 - F_{|h_2|^2}(\Theta_2 x^\delta)] dx \\[2mm]
&= 1 - \frac{2\mu_2^{m_2} \Theta_2^{m_2}}{m_2!(R_2^2 - R_1^2)} \int_{R_1}^{R_2} x^{\delta m_2 + 1} dx \\[2mm]
&= 1 - \frac{2\mu_2^{m_2} \Theta_2^{m_2}}{m_2!(R_2^2 - R_1^2)(\delta m_2 + 2)}\left[R_2^{\delta m_2 + 2} - R_1^{\delta m_2 + 2}\right].
\end{aligned}
\tag{26}
$$

Finally, we obtain the asymptotic outage probabilities of $D_1$ and $D_2$ from (27) and (28), respectively:

$$
\begin{aligned}
\mathcal{P}_{out}^{1,\infty} &= 1 - \left\langle 1 - \frac{1}{\Gamma(\alpha)\Gamma(\beta)}\left[\frac{\Gamma(\beta - \alpha)}{\alpha}(\xi \Theta_{\max})^\alpha + \frac{\Gamma(\alpha - \beta)}{\beta}(\xi \Theta_{\max})^\beta\right]\right\rangle \\[2mm]
&\times \left[1 - 2\frac{\mu_1^{m_1} \Theta_{\max}^{m_1} R_1^{m_1 \delta}}{m_1!(m_1 \delta + 2)}\right].
\end{aligned}
\tag{27}
$$

$$
\begin{aligned}
\mathcal{P}_{out}^{2,\infty} &= 1 - \left\langle 1 - \frac{1}{\Gamma(\alpha)\Gamma(\beta)}\left[\frac{\Gamma(\beta - \alpha)(\xi \Theta_{\max})^\alpha}{\alpha} + \frac{\Gamma(\alpha - \beta)(\xi \Theta_{\max})^\beta}{\beta}\right]\right\rangle \\[2mm]
&+ \left\langle 1 - \frac{1}{\Gamma(\alpha)\Gamma(\beta)}\left[\frac{\Gamma(\beta - \alpha)(\xi \Theta_{\max})^\alpha}{\alpha} + \frac{\Gamma(\alpha - \beta)(\xi \Theta_{\max})^\beta}{\beta}\right]\right\rangle \\[2mm]
&\times \frac{2\mu_2^{m_2} \Theta_2^{m_2}[R_2^{\delta m_2 + 2} - R_1^{\delta m_2 + 2}]}{m_2!(R_2^2 - R_1^2)(\delta m_2 + 2)}.
\end{aligned}
\tag{28}
$$

## 5 Numerical results

In this section, we evaluate the performance of the derived theoretical expression and also validate it with numerical results. The fading parameters were set to $m = m_{h_0} = m_{h_1} = m_{h_2}$. Table 3 summarizes the main parameters. Furthermore, the corresponding noise power at $D_1$ and $D_2$ was computed as $\sigma^2 = N_0 + 10 \log (BW) + NF$ [dBm] in [65]. The technological advancement of our code resides in the implementation of symbolic computations within Matlab, which facilitated the attainment of highly precise results.

**Table 3. Main parameters for the simulations [55, 62].**

| Parameters | Notation | Values |
|---|---|---|
| Power splitting factors | $\{a_1, a_2\}$ | $\{0.1, 0.9\}$ |
| Target rates decode $x_1$ and $x_2$ | $\{R_{th}^1, R_{th}^2\}$ | $\{1, 0.5\}$ |
| Inner circle radius | $R_1$ | 50 m |
| Outer circle radius | $R_2$ | 100 m |
| Path-loss exponent | $\delta$ | 3 |
| Channel gains | $\{\lambda_1, \lambda_2\}$ | $\{0.1, 0.1\}$ |
| Fading severity parameter | $m$ | 3 |
| Bandwidth | BW | 10 [MHz] |
| Noise figure | NF | 10 [dBm] |
| Thermal noise power density | $N_0$ | −174 [dBm/Hz] |
| SIC efficiency | $\chi$ | 0.01 |
| Distance between $S$ and $R$ | $L$ | 4000 m |
| Receiver aperture diameter | $D$ | 0.01 m |
| Operational wavelength | $\lambda$ | 1550 nm |
| Turbulence strength | $C_n^2$ | $3 \times 10^{-14} \mathrm{m}^{-2/3}$ |

Fig 2 illustrates the outage probability of a dual-hop FSO/RF system as a function of the SIC capability, denoted $\chi$, for near users. The outage probability decreases as the value of $m$ increases from 1 to 3 to 5. This trend can be explained by a larger value of $m$ resulting in a larger sequence containing the sum function in Eq (19), leading to a smaller OP when 1 minus

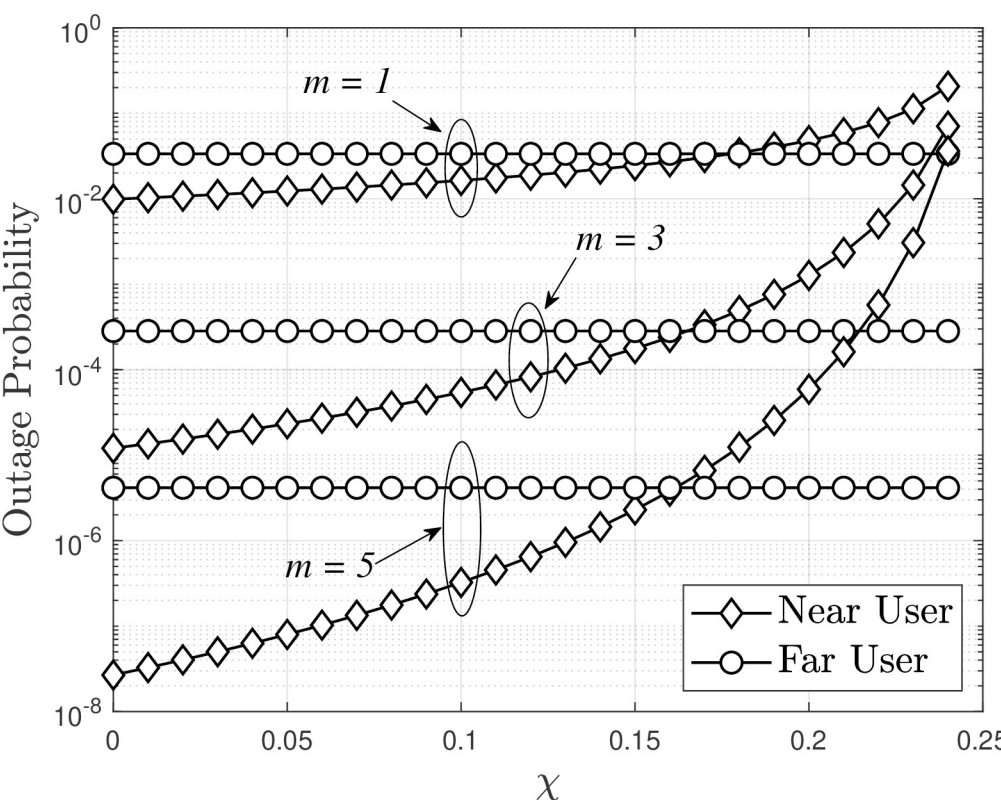

**Fig 2. Outage probability of a dual-hop FSO/RF system versus SIC capability $\chi$, with $R_{th}^1 = R_{th}^2 = 0.5$, $a_1 = 0.2$, $a_2 = 0.8$ and $P_S = -10$ [dBm].**

that series is considered. As the SIC capability $\chi$ changes from perfect SIC ($\chi = 0$) to $\chi = 0.25$, the OP also varies, reaching its lowest value at $\chi = 0$ and gradually increasing as $\chi$ increases. Notably, when $\chi$ is approximately 0.24, the OP approaches an asymptotic value for all three cases of $m$ considered. This observation highlights that the more effective the relay's effective interference cancellation capability, the better the system's performance.

In Fig 3, the OP of the system is evaluated as a function of $P_S$(dBm). It is observed that an increase in $P_S$(dBm) leads to a decrease in the OP. This is because higher $P_S$(dBm) values result in higher energy at the relay, which significantly improves the received SINR. In addition, increasing $m$ results in better OP performance, similar to the trend observed in Fig 2. The asymptotic OP of destinations $D_1$ and $D_2$, as presented in Eqs (27) and (28), is also accurately simulated in Fig 3. Specifically, for $P_S > -15$ dBm, the asymptotic OP and the actual OP are almost identical.

In addition to increasing $m$ in order to improve OP performance, as observed in Figs 2 and 4 indicates that two users experience different outage performance as the power allocation factor varies. Specifically in this case, the lowest OP for user $D_1$ occurs when $a_2 = 0.6$, while for user $D_2$, it occurs when $a_2 = 0.95$. This behavior can be explained by considering that as $a_2$ is increases, $a_1$ decreases, leading to an increase in the received SINR at $D_2$, as indicated by Eq (7). Hence, the OP at the $D_2$ source always decreases. However, the received SINR at $D_1$ may increase (based on Eq (5)) or decrease (based on Eq (6)). Therefore, the OP expression in Eq (18) does not show a clear trend of increasing or decreasing, but through simulation and under the condition $a_2 > a_1$, when $a_2$ increases from 0.6 to 0.95, the OP of near users also increases and converges to an asymptotic value when $a_2 = 0.95$. As a result, as $a_2$ increases, the

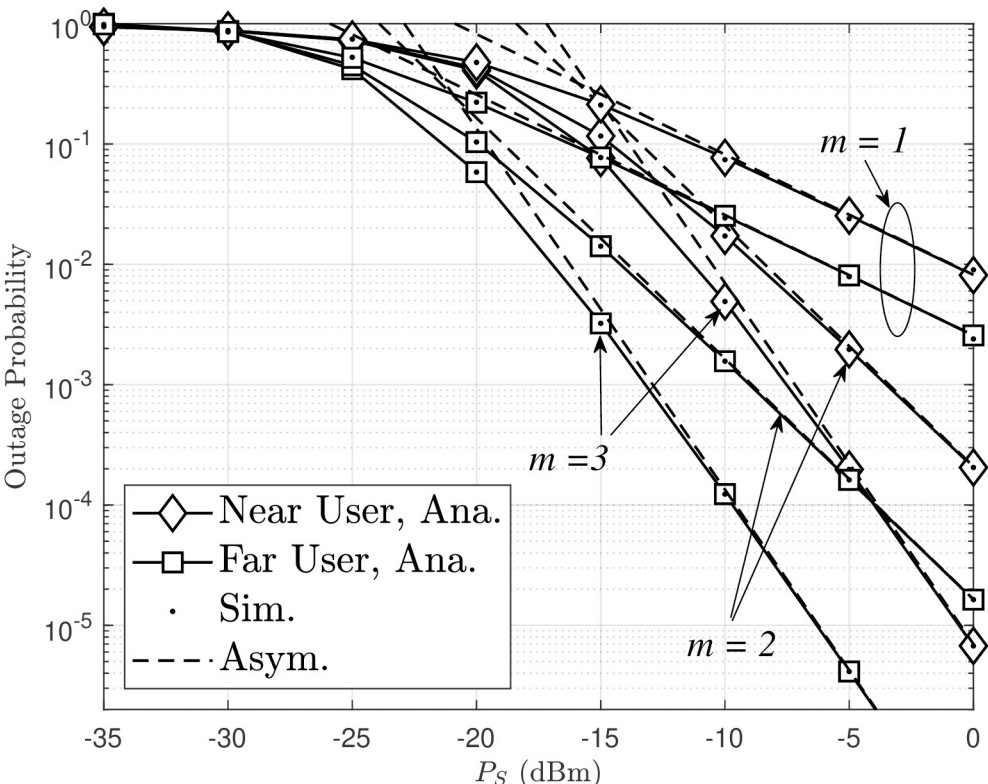

**Fig 3. Outage probability of a dual-hop FSO/RF system versus $P_S$.**

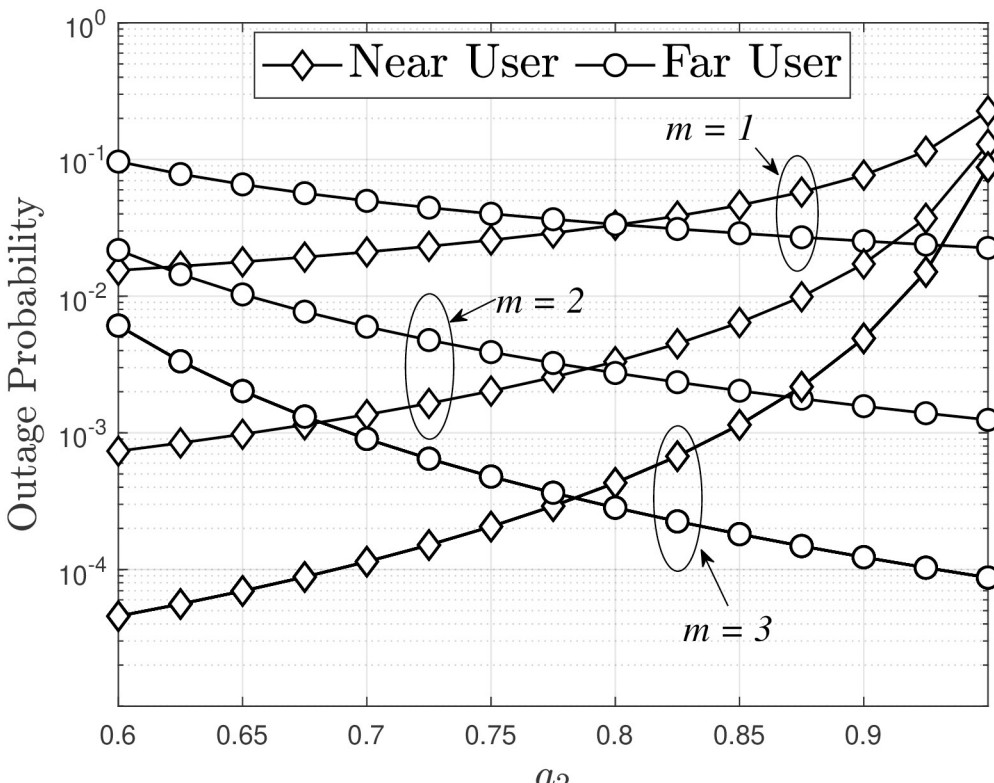

**Fig 4. Outage probability versus power allocation factors, with $P_S = -10$ [dBm], $R_{th}^1 = 1$ and $R_{th}^2 = 0.5$.**

far user's OP performance improves while the near user's OP performance deteriorates, and vice versa. To ensure fairness between users, an optimal level of the power allocation factor exists; this is achieved when $a_2 \approx 0.7$ (corresponding to $a_1 = 0.3$), resulting in nearly equal OP performance for both near and far users.

Fig 5 illustrates the outage probability of the model when compared to $R_{th}^1 = R_{th}^2$, with $\rho = 30$ [dB], $a_1 = 0.05$ and $a_2 = 0.95$. As $R_{th}^1 = R_{th}^2$ increases, the user's OP efficiency decreases, as explained by Eqs (18) and (20). In addition, the OP performance of distant users is better than that of proximity users. This is because, with a power allocation factor $a_2 = 0.95$, distant users achieve their best OP performance (as previously indicated in Fig 4). In Fig 6 presents the outage probability versus PS (in dBm) for different scenarios in a wireless communication system, showing that the NOMA scheme achieves better performance for both near and far users compared to the conventional OMA, with asymptotic and simulated results depicted by dashed and solid lines, respectively, demonstrating the superior robustness of NOMA, especially at higher transmit power levels.

Fig 7 shows the outage probability of FSO and FSO-RF links at both near and far users as a function of $P_S$(dBm), with simulated (solid lines) and asymptotic (dashed lines) results indicating better performance at higher power levels. the figure highlights that both user distance and link type (FSO or FSO-RF) significantly impact outage probability, with the near-user FSO link providing the best reliability. Increasing transmit power $P_S$ reduces the outage probability across all configurations, the performance of users for FSO-RF NOMA network is better than FSO NOMA network.

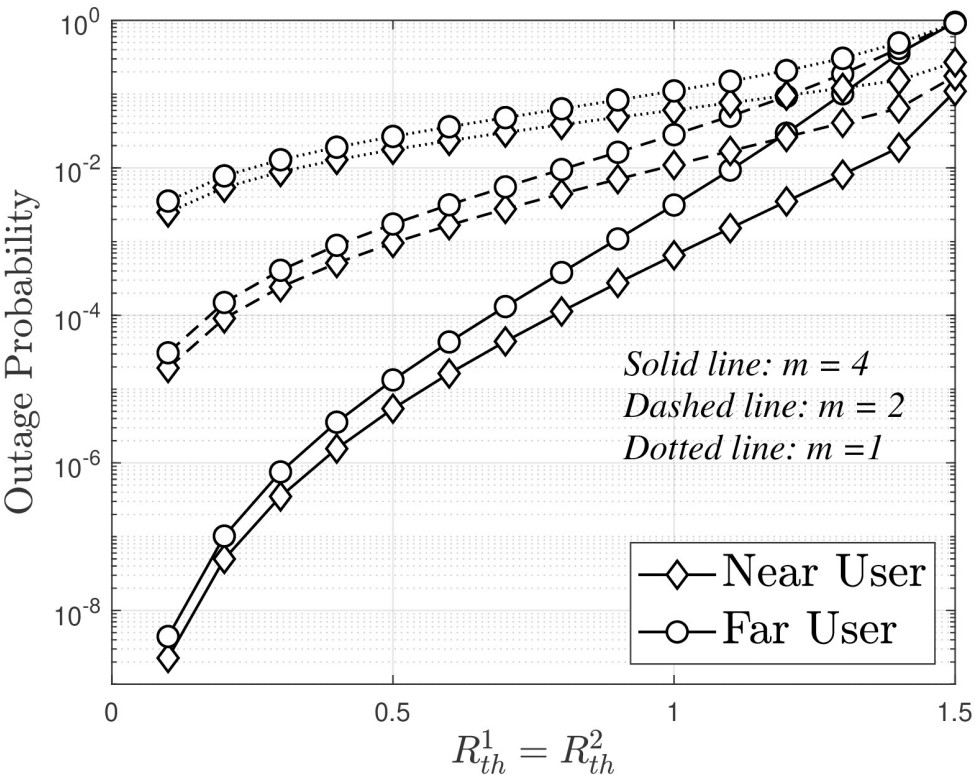

**Fig 5. Outage probability versus** $R_{th}^1 = R_{th}^2$**, with** $P_S = -10$ **[dBm],** $a_1 = 0.12$ **and** $a_2 = 0.88$**.**

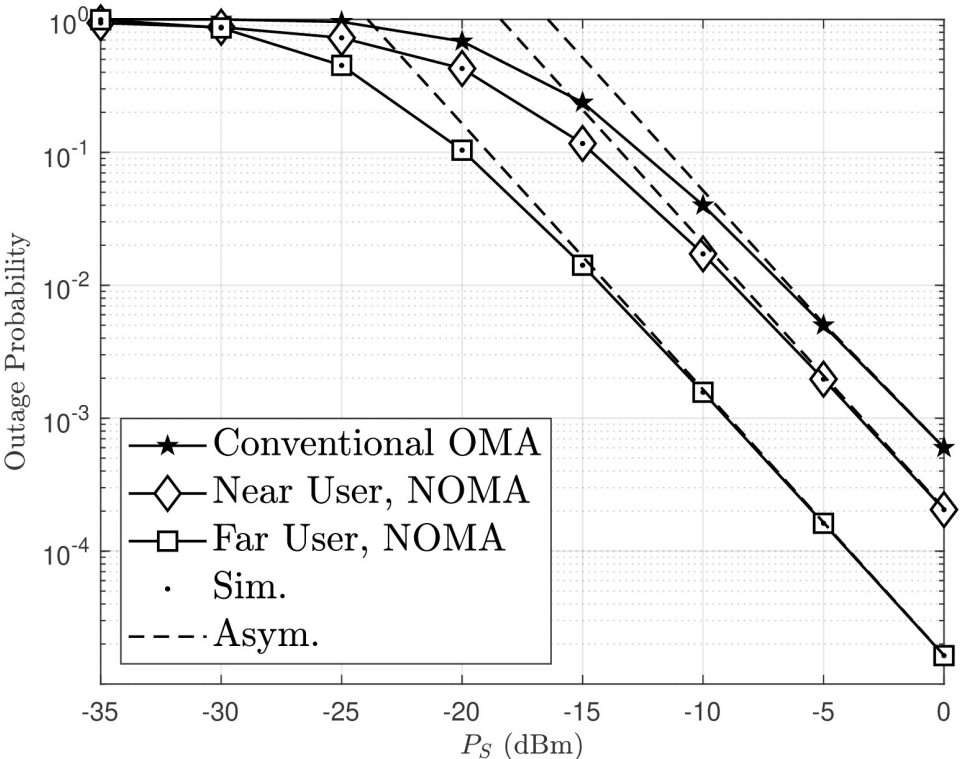

**Fig 6. Outage probability versus** $P_S$ **in NOMA and OMA, with** $m = 2$**,** $a_1 = 0.1$ **and** $a_2 = 0.9$**.**

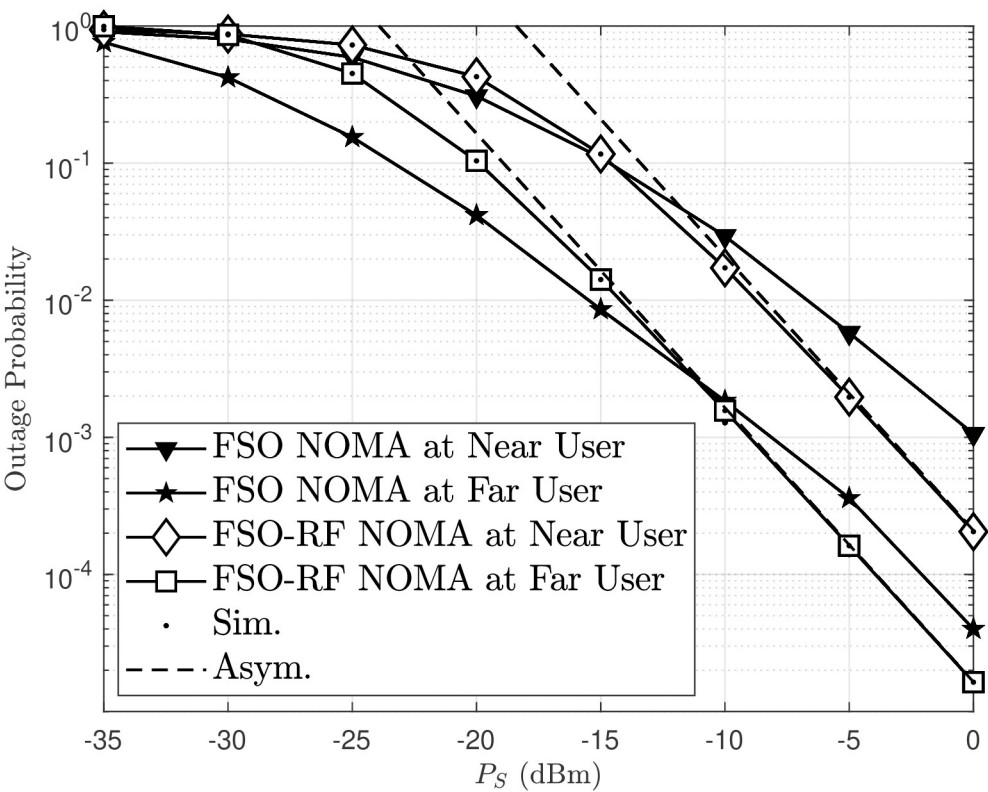

**Fig 7. Outage probability versus $P_S$ in FSO link and FSO-RF link, with $m = 2$ and $\chi = 0.01$.**

## 6 Conclusion

This paper provides a performance analysis of a dual-hop mixed RF-FSO system integrated with NOMA communications technology. Using DF relaying, the relay R employs successive interference cancellation, where the FSO link from the source to the relay follows a double Gamma distribution, and the RF link from the relay to multiple users follow the Nakagami-$m$ distribution. Based on this system model, we analyzed the outage probability. Our results demonstrate that SIC has a clear effect on OP performance; specifically the greater the SIC capability, the better the system's performance. We also varied the Nakagami-$m$ channel parameters to examine their effect on system performance. The results indicate that increasing the $m$ value enhances system performance. The power allocation factors also notably affect OP performance for both near and far users. A higher power allocation factor for far users results in improved OP efficiency for those users but reduced OP efficiency for near users, and vice versa. Consequently, there is a trade-off in OP performance between users. However, an optimal level of the power allocation factor exists that ensures fairness between users. The system's performance also depends on other system parameters. An analysis of these parameters provides insight into optimizing the design and operation of dual-hop mixed RF-FSO systems with NOMA technology.

## Supporting information

**S1 Appendix. Proof of Proposition 1.**
(PDF)

**S2 Appendix. Proof of Proposition 2.**
(PDF)

**S1 File.**
(RAR)

## Author Contributions

**Data curation:** Thu-Ha Thi Pham.

**Formal analysis:** Tran Cong Hung.

**Funding acquisition:** Miroslav Voznak.

**Investigation:** Tran Cong Hung, Pham Ngoc Son, Miroslav Voznak.

**Methodology:** Tran Cong Hung, N. H. K. Nhan.

**Software:** Tan N. Nguyen, Anh-Tu Le, Thu-Ha Thi Pham.

**Validation:** Tan N. Nguyen.

**Writing – review & editing:** N. H. K. Nhan, Pham Ngoc Son.

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
