## [Decision Letter · Decision Letter 0]

4 Oct 2024

PONE-D-24-24546Performance analysis of dual-hop mixed RF-FSO systems combined with NOMAPLOS ONE

Dear Dr. Le,

Thank you for submitting your manuscript to PLOS ONE. After careful consideration, we feel that it has merit but does not fully meet PLOS ONE’s publication criteria as it currently stands. Therefore, we invite you to submit a revised version of the manuscript that addresses the points raised during the review process.

We look forward to receiving your revised manuscript.

Kind regards,

Rahat Ullah, Ph.D

Academic Editor

PLOS ONE

Journal Requirements: When submitting your revision, we need you to address these additional requirements. 1. Please ensure that your manuscript meets PLOS ONE's style requirements, including those for file naming. The PLOS ONE style templates can be found at https://journals.plos.org/plosone/s/file?id=wjVg/PLOSOne_formatting_sample_main_body.pdf and https://journals.plos.org/plosone/s/file?id=ba62/PLOSOne_formatting_sample_title_authors_affiliations.pdf 2. Please note that PLOS ONE has specific guidelines on code sharing for submissions in which author-generated code underpins the findings in the manuscript. In these cases, all author-generated code must be made available without restrictions upon publication of the work. Please review our guidelines at https://journals.plos.org/plosone/s/materials-and-software-sharing#loc-sharing-code and ensure that your code is shared in a way that follows best practice and facilitates reproducibility and reuse. 3. We note that the grant information you provided in the ‘Funding Information’ and ‘Financial Disclosure’ sections do not match.  When you resubmit, please ensure that you provide the correct grant numbers for the awards you received for your study in the ‘Funding Information’ section. 4. We note that your Data Availability Statement is currently as follows: All relevant data are within the manuscript and its Supporting Information files Please confirm at this time whether or not your submission contains all raw data required to replicate the results of your study. Authors must share the “minimal data set” for their submission. PLOS defines the minimal data set to consist of the data required to replicate all study findings reported in the article, as well as related metadata and methods (https://journals.plos.org/plosone/s/data-availability#loc-minimal-data-set-definition). For example, authors should submit the following data: - The values behind the means, standard deviations and other measures reported;- The values used to build graphs;- The points extracted from images for analysis. Authors do not need to submit their entire data set if only a portion of the data was used in the reported study. If your submission does not contain these data, please either upload them as Supporting Information files or deposit them to a stable, public repository and provide us with the relevant URLs, DOIs, or accession numbers. For a list of recommended repositories, please see https://journals.plos.org/plosone/s/recommended-repositories. If there are ethical or legal restrictions on sharing a de-identified data set, please explain them in detail (e.g., data contain potentially sensitive information, data are owned by a third-party organization, etc.) and who has imposed them (e.g., an ethics committee). Please also provide contact information for a data access committee, ethics committee, or other institutional body to which data requests may be sent. If data are owned by a third party, please indicate how others may request data access. 5. Please amend either the abstract on the online submission form (via Edit Submission) or the abstract in the manuscript so that they are identical. 6. Please ensure that you refer to Figure 1 in your text as, if accepted, production will need this reference to link the reader to the figure. 7. We note you have included a table to which you do not refer in the text of your manuscript. Please ensure that you refer to Table 1 and 2 in your text; if accepted, production will need this reference to link the reader to the Table. 8. We notice that your supplementary [figures/tables] are included in the manuscript file. Please remove them and upload them with the file type 'Supporting Information'. Please ensure that each Supporting Information file has a legend listed in the manuscript after the references list.

Reviewers' comments:

Reviewer's Responses to Questions

**Comments to the Author**

1. Is the manuscript technically sound, and do the data support the conclusions?

Reviewer #1: Yes

Reviewer #2: Yes

2. Has the statistical analysis been performed appropriately and rigorously? 

Reviewer #1: Yes

Reviewer #2: Yes

3. Have the authors made all data underlying the findings in their manuscript fully available?

Reviewer #1: Yes

Reviewer #2: No

4. Is the manuscript presented in an intelligible fashion and written in standard English?

Reviewer #1: Yes

Reviewer #2: Yes

5. Review Comments to the Author

Reviewer #1: This study considered the Performance Analysis of Dual-Hop Mixed RF-FSO Systems Combined with NOMA. In general, this paper is well-written, and I thoroughly enjoyed reading it. It has a limitation of contribution compared with previous papers. Some weak points need to be addressed

1. In this paper, the author considers two users. How about multiple users?

2. The authors should compare your work with the case of OMA.

3. How do the power allocation factors influence the system performance of the considered system

4. What is the CSI assumption of the article, please describe it in a real scenario.

5. The presentation of the whole paper is poor. We need more improvements.

Reviewer #2: The introduction is so general. The topic of the paper is about FSO and NOMA. The general discussion should be removed, and the related literature should be included. The current challenges should be cited from the papers, which the authors would address in the rest of the paper. The given references could also be helpful:

- https://opg.optica.org/ol/fulltext.cfm?uri=ol-48-17-4548&id=536662

- https://www.mdpi.com/2304-6732/10/10/1073

- https://opg.optica.org/ol/abstract.cfm?uri=ol-48-15-4101

- https://www.sciencedirect.com/science/article/abs/pii/S1434841123005587?via%3Dihub

B: Many of the acronyms are not repeated in the abstract, so it is better to remove them.

C: Details about NOMA and its working principal is missing. I) how will it work in FSO system, ii) what will be/could be the transmission range.

D: It is strongly claimed by the authors: This paper provides a performance analysis of a dual-hop mixed RF-FSO system integrated with NOMA communications technology. While in the paper after section 2, I don’t see any evidence/contribution of the authors about NOMA, they have discussed numerically FSO and RF in details, but nothing is shown/presented about NOMA.

D: How does the Hybrid FSO+NOMA work? i) this point is not discussed, ii) similarly, the numerical+theoretical details are missing, and iii) results related to this claim are not presented clearly. I would suggest the authors to revise the paper in detail.

6. PLOS authors have the option to publish the peer review history of their article (what does this mean?). If published, this will include your full peer review and any attached files.

Reviewer #1: No

Reviewer #2: No

---

## [Author Response · Author response to Decision Letter 0]

4 Nov 2024

Dear Reviewers 

We would like to thank you so much for your review of our manuscript. We have attached the response letter in the submitted manuscript to show our response to your comments. Thank you

Best regards

---

## [Decision Letter · Decision Letter 1]

21 Nov 2024

Performance Analysis of Dual-Hop Mixed RF-FSO Systems Combined with NOMA

PONE-D-24-24546R1

Dear Dr. Le,

We’re pleased to inform you that your manuscript has been judged scientifically suitable for publication and will be formally accepted for publication once it meets all outstanding technical requirements.

Kind regards,

Rahat Ullah, Ph.D

Academic Editor

PLOS ONE

Additional Editor Comments (optional):

I have no more comments.

Reviewers' comments:

Reviewer's Responses to Questions

Reviewer #1: All comments have been addressed

2. Is the manuscript technically sound, and do the data support the conclusions?

Reviewer #1: Yes

3. Has the statistical analysis been performed appropriately and rigorously? 

Reviewer #1: Yes

4. Have the authors made all data underlying the findings in their manuscript fully available?

Reviewer #1: No

5. Is the manuscript presented in an intelligible fashion and written in standard English?

Reviewer #1: Yes

6. Review Comments to the Author

Reviewer #1: (No Response)

7. PLOS authors have the option to publish the peer review history of their article (what does this mean?). If published, this will include your full peer review and any attached files.

Reviewer #1: No

---

## [Editor Report · Acceptance letter]

8 Dec 2024

PONE-D-24-24546R1 

PLOS ONE

Dear Dr. Le, 

I'm pleased to inform you that your manuscript has been deemed suitable for publication in PLOS ONE. Congratulations! Your manuscript is now being handed over to our production team.

Kind regards, 

on behalf of

Dr. Rahat Ullah 

Academic Editor

PLOS ONE